# Profiling of Myositis Specific Antibodies and Composite Scores as an Aid in the Differential Diagnosis of Autoimmune Myopathies

**DOI:** 10.3390/diagnostics11122246

**Published:** 2021-11-30

**Authors:** Michael Mahler, Kishore Malyavantham, Andrea Seaman, Chelsea Bentow, Ariadna Anunciacion-Llunell, María Teresa Sanz-Martínez, Laura Viñas-Gimenez, Albert Selva-O’Callaghan

**Affiliations:** 1Research and Development, Inova Diagnostics, San Diego, CA 92131, USA; kmalyavantha@inovadx.com (K.M.); aseaman@werfen.com (A.S.); cbentow@werfen.com (C.B.); 2Autoimmune Systemic Diseases Unit, Department of Internal Medicine, Hospital Vall d’Hebron, Universitat Autònoma de Barcelona, 08035 Barcelona, Spain; ariadna.anunciacion@vhir.org (A.A.-L.); aselva@vhebron.net (A.S.-O.); 3Immunology Department, Hospital Vall d’Hebrón, Universitat Autònoma de Barcelona, 08035 Barcelona, Spain; mtsanzmartinez@gmail.com (M.T.S.-M.); lauravinasgi@gmail.com (L.V.-G.)

**Keywords:** myositis, autoantibodies, diagnosis, polymyositis, dermatomyositis, immunoassay

## Abstract

(1) Background: Myositis specific antibodies (MSA) represent important diagnostic and stratification tools in idiopathic inflammatory myositis (IIM) patients. Here we aimed to evaluate the clinical performance of MSA profiled by a novel particle based multi-analyte technology (PMAT) in IIM and subsets thereof. (2) Methods: 264 IIM patients and 200 controls were tested for MSA using PMAT (Inova Diagnostics, research use only). Diagnostic performance was analyzed and composite scores were generated. (3) Results: The sensitivity/specificity of the individual MSA were: 19.7%/100% (Jo-1), 7.2%/100.0% (Mi-2), 3.0%/99.0% (NXP2), 3.8%/100.0% (SAE), 2.7%/100.0% (PL-7), 1.9%/99.5 (PL-12), 1.1%/100.0% (EJ), 15.5%/99.5% (TIF1γ), 8.3%/98.5% (MDA5), 6.1%/99.0% (HMGCR) and 1.9%/98.5% (SRP). Of all IIM patients, 180/264 tested positive for at least one of the MSAs. In the individual control group, 12/200 (6.0%) tested positive for at least one MSA, most of which had levels close to the cut-off (except one SRP and one PL-12). Only 6/264 (2.3%) IIM patients were positive for more than one antibody (MDA5/HMGCR, EJ/PL-7, 2 x MDA5/TIF1γ, EJ/SAE, SAE/TIF1γ). The overall sensitivity was 68.2% paired with a specificity of 94.0%, leading to an odds ratio of 33.8. The composite scores showed good discrimination between subgroups (e.g., anti-synthetase syndrome). (4) Conclusion: MSA, especially when combined in composite scores (here measured by PMAT), provide value in stratification of patients with IIM.

## 1. Introduction

Myositis specific (MSA) and myositis associated antibodies (MAA) have been used as an aid in the diagnosis of idiopathic inflammatory myopathies (IIM) for decades [1]. Since many of the MSA (e.g., anti-synthetase antibodies), partly depending on the screening dilution, are accompanied by limited sensitivity of the indirect immunofluorescence (IIF) test on HEp-2 cells [2,3,4,5], confirmatory tests are used at screening level. At present, besides immunoprecipitation (IP), mostly line immunoassays (LIA) and dot blot (DB) assays are routinely used for the detection of MSA [6], which are convenient tools for the simultaneous detection of various antibodies, but are also accompanied by some limitations [7] including the lack of true quality controls [8], lack of sensitivity [9,10] and specificity (10) for some analytes and subjectivity in interpretation [10,11,12].

Especially during the past 10–15 years, many novel and clinically relevant MSA have been identified [1] which can aid in the diagnosis and stratification of IIM. Since publication of updated new classification criteria for IIM [13,14], a debate has been triggered about the omission of MSA (except anti-Jo-1 antibodies), which was eventually explained by the lack of standardization of autoantibody assays and missing data derived from large multi-centric studies [15,16]. In addition, an alternative classification approach has been proposed that leverages both clinical and autoantibody data [17,18]. For the reasons summarized above, new technologies and methods for the detection of MSA are warranted and proper validation is needed. In addition, the large and evolving number of MSA often complicates the interpretation of test results, especially for clinicians not too specialized in IIM. The aim of the present study was to evaluate the diagnostic performance of MSA for IIM and in particular for IIM subsets as well as using composite scores derived from a novel particle-based multi-analyte technology (PMAT).

## 2. Materials and Methods

### 2.1. Patient Cohort

The study included 464 serum samples collected from patients at Hospital Vall d’Hebron, University of Barcelona, 264 of whom had a diagnosis of IIM. Of those, 67 were classified as anti-synthetase syndrome (ASS), 70 as dermatomyositis (DM), 35 as cancer associated DM (cDM), 21 as clinically amyopathic DM (CADM), 16 as immune-mediated necrotizing myopathies (IMNM), six as inclusion body myositis (IBM), 20 as overlap syndrome (OS), five as juvenile DM (JDM) and 23 as polymyositis (PM). One patient with elevated CK levels and muscle weakness and interstitial lung disease was included as IIM, but did not have a final diagnosis at the time point of the study (they were excluded from subtype analysis). Part of the cohort was described and analyzed in a previous study [19]. Controls included samples from patients with myositis-like conditions (ML, n = 20), rheumatoid arthritis (RA, n = 33), systemic lupus erythematosus (SLE, n = 40), Sjögren‘s syndrome (SjS, n = 25), infectious diseases (ID, n = 40) and healthy individuals (HI, n = 42). Diagnoses were established in concordance with the respective and recognized disease classification criteria. The study was approved by the Ethics Commission of Hospital Vall d’Hebron, Barcelona (Spain) (PR (AG) 223/2013). For details about the patient cohort see supplement materials.

### 2.2. Autoantibody Testing

All samples were tested using a novel fully automated particle-based multi-analyte technology (PMAT, Inova Diagnostics, research use only; Jo-1, PL-7, PL-12, EJ, Mi-2ß, TIF1γ, SAE, MDA5, NXP2, HMGCR, SRP) which utilizes paramagnetic particles with unique signatures and a digital interpretation system as described previously [20,21,22]. In brief: antigens are coupled to paramagnetic particles that carry unique signatures and incubated with diluted patient samples. After 9.5 min incubation at 37 °C, particles are washed and incubated 9.5 min, then incubated at 37 ºC with anti-human IgG conjugated to phycoerythrin (PE) to label the bound autoantibodies. After the final wash cycle, median fluorescence intensity (MFI) on the particles is captured using a digital imager and analyzed using proprietary algorithms to extract meaningful information for each analyte. Cut-off values for each analyte were previously established using IIM patients (n > 250), as well as healthy and disease controls (n = 840) using receiver operating characteristic (ROC) analysis, as previously outlined [21,22].

### 2.3. Statistical Methods

Results derived from PMAT for each analyte were normalized by dividing the MFI value of the samples by the respective cut-off. Receiver operating characteristic (ROC) curve analysis with area under the curve (AUC), scatter plots, sensitivity, specificity, odds ratio values in the context of IIM (n = 264) vs. controls (n = 200) and within each IIM subtype (calculated using 264 IIM samples only) were calculated using Analyse-it software (Leeds, UK). The control group did not contain antibody-positive individuals for some antibody combinations, which interferes with the calculation of odds ratio (OR) and positive likelihood ratio (LR+). To estimate these values, a pseudo-frequency modification was used. This modification entails adding a small number to each cell in the contingency table. For replacing ‘0’s in the contingency tables, 0.2 was used when % positivity is <10% and 0.5 when % positivity is >10% following the approach described previously [23]. Visualization of the overlap in positivity for various MSA in IIM subtypes was accomplished using the ‘Upset plots’ library in Python as descried by Lex and colleagues [24]. Sensitivity and specificity of each MSA were calculated in two different ways: IIM cases vs non-IIM controls (RA, ML, SLE, SjS, ID and HI); and particular IIM subtype in comparison to the remaining 264 IIM cases that composed this cohort. Composite scores (MyoScores) were generated based on all MSA tested (sum of MFI reactivities).

## 3. Results

### 3.1. Prevalence of Myositis Specific Antibodies in Myositis and Controls

The sensitivity/specificity of the individual MSA were: 19.7%/100% (Jo-1), 7.2%/100.0% (Mi-2), 3.0%/99.0% (NXP2), 3.8%/100.0% (SAE), 2.7%/100.0% (PL-7), 1.9%/99.5 (PL-12), 1.1%/100.0% (EJ), 15.5%/99.5% (TIF1γ), 8.3%/98.5% (MDA5), 6.1%/99.0% (HMGCR) and 1.9%/98.5% (SRP) (Table 1, Figure 1). 

When the diagnostic performance was studied by ROC analysis, AUC values were 0.612 (Jo-1), 0.59 (EJ), 0.621 (PL-7), 0.640 (PL-12), Mi-2ß (0.637), 0.787 (TIF1γ), 0.632 (SAE), NXP2 (0.576), 0.623 (MDA5), 0.513 (SRP) and 0.574 (HMGCR) (see Figure 2). Although the discrimination for most analytes was limited based on the AUC values, in the relevant area of the ROC curve (high specificity), most markers showed discrimination, albeit with low sensitivity. Of all IIM patients, 180/264 (68.2%) tested positive for at least one of the MSA. In the individual control groups, 0/20 (0.0%) of ML, 2/33 (6.1%) of RA, 5/40 (12.5%) of SLE, 2/25 (8.0%) of SjS, 2/40 (5.0%) of ID and 1/42 (2.4%) of HI were positive for at least one MSA (Figure 1, Appendix A) resulting in 94.0% overall specificity. Among the control samples that were positive, 8/11 had low levels (< 3 x cut-off). The three controls samples with high levels of MSA were positive for PL-12 (10.2 x cut-off), MDA5 (4.1 x cut-off) and SRP (7.5 x cut-off). In the IIM group, only 6/264 (2.3%) patients were positive for more than one antibody (MDA5/HMGCR, EJ/PL-7, 2 − MDA5/TIF1y, EJ/SAE, SAE/TIF1y) (Appendix A). The co-existence of autoantibodies is described using an ‘UpSet plot’ in Figure 1d. When the sensitivity and specificity of the MSA were analyzed within the associated clinical subsets (e.g., Jo-1 for ASS), the diagnostic performance significantly increased for most of the MSA (see Figure 1b and Table 1). 

### 3.2. Quantitative Analysis of Myositis Specific Antibodies Studied

The relative level of positivity for tested MSA were depicted on a common y-axis (Figure 1) for both IIM and controls, by using normalized values generated by dividing the MFI of the respective sample by the cut-off value for the respective MSA. For all MSA, the levels of antibodies were significantly higher in IIM vs. controls (*p* < 0.05). The highest antibody levels were observed for PL-7 and MDA5 with values up to 22 and 30-fold of the cut-off values, respectively. 

### 3.3. Prevalence of Myositis Specific Antibodies in Myositis Subsets

When the individual MSA were evaluated in their respective IIM subsets, a significantly higher prevalence was observed for each: E.g. the prevalence for Jo-1, PL-7, PL-12 and EJ within the ASS subset was 71.6%, 10.4%, 7.5% and 3.0%, respectively (Table 2). 

When analyzing the MSA associated with DM, prevalences of 21.4% (Mi-2ß), 7.1% (NXP-2), 11.4% SAE and 15.7% (TIF1γ) were observed. TIF1γ had a prevalence of 62.9% in cDM and MDA5 71.4% in CADM. Lastly, HMGCR and SRP were present in 50.0% and 12.5% of IMNM patients, respectively (see Table 2). The cluster plots (Figure 1b,c) illustrate the association with the IIM subset. In addition, the odds ratios for the individual MSA showed significant differences, some showing much higher values comparing IIM subsets (see Figure 3).

### 3.4. Generation and Performance Assessment of MyoScores for Idiopathic Inflammatory Myopathies

Next, we generated composite scores (MyoScores) using the results of all MSA tested and used the numerical outcome for performance statistics. Receiver operating characteristic (ROC) analysis showed good discrimination between IIM patients and controls with an AUC of 0.87 (0.83–0.90) (Figure 3). The overall diagnostic performance was: sensitivity 68.2% (95% confidence interval 62.3–73.5%), specificity 94.0% 95% CI 89.8–96.5) and odds ratio 33.8. The sensitivity and specificity of the MyoScore were used to generate pre-test/post-test probability plots that can be used to estimate the post-test probability of a patient based on the clinical features and the test result of the MyoScore (see Figure 3).

When Jo-1, PL-7, PL-12, EJ were combined in a score for ASS, very good discrimination was observed with an AUC in ROC of 0.94 (Figure 4). MDA5 by itself yielded an AUC of 0.80 for CADM and while TIF1γ provide an AUC of 0.81 for cDM. HMGCR and SRP combined in a score resulted in an AUC of 0.80 for IMNM. Lastly, Mi-2ß, NXP2, SAE, TIF1γ gave an AUC of 0.73 for DM. 

## 4. Discussion

Despite the expansion of MSA as an aid in the diagnosis and stratification of IIM and their global adoption, significant challenges persist, which include the lack of standardization and the difficulty in interpretation of results. Careful validation of autoantibody assays for the detection of MSA is important, since some of these antibodies are included or being considered for IIM classification criteria [1,2,3,4,5,6,7,8,9,10,11,12,13,14,15,16,17,18,19,20,21,22,23,24,25,26]. Although several other MSA also carry clinical utility, only anti-Jo-1 antibodies have been included in the recent EULAR/ACR classification criteria for IIM, mostly due to the lack of data and standardization of other MSA. In addition, several autoantibodies showed relevance for a novel approach to classify IIM [17,18,19,20,21,22,23,24,25,26,27]. Consequently, the markers are not only relevant as an aid in the diagnosis, but also in the stratification to specific disease subsets [1,2,3,4,5,6,7,8,9,10,11,12,13,14,15,16,17,18,19,20,21,22,23,24,25,26]. Most of the clinical associations of MSA and MAA have been established using IP and might be lost with newer methods [16,17,18,19,20,21,22,23,24,25,26,27,28]. 

According to a recent survey, when analyzed globally, about 35% of laboratories use ELISA and another ~35% LIA for the routine detection of MSA [6,25]. The remaining percentage is fragmented between immunoprecipitation (IP), laser bead assays and other methods. However, it needs to be acknowledged that not many MSA can be detected using commercially available ELISA. Another limitation of the data derived from the survey is the number of samples processed in each responding laboratory. This is of particular importance, since myositis testing is often done in large reference laboratories (depending on the geography). Consequently, the results from the survey might be biased. However, it is evident that LIA or dot blots (DB) have become convenient tools for the simultaneous detection of various antibodies, and some of the markers contained on LIA show good to excellent correlation with IP [25]. However, LIA and DB are also accompanied by some limitations, including the lack of true quality controls [8,25] and the lack of sensitivity or specificity for some analytes [9,10,11,12,13,14,15,16,17,18,19,20,21,22,23,24,25,26,27,28,29,30]. Recently, data from different studies were summarized showing varying agreement (from low to high, kappa: PM/Scl, 0.48; Jo-1, 0.52; Mi-2; 0.56; SRP, 0.60; TIF1y, 0.71; PL-12, 0.72; EJ, 0.72; MDA5, 0.75; SAE, 0.79; Ku, 0.83; NXP-2, 0.86; PL7, 0.82). Part of this inter-method variability might be attributed to the subjectivity associated with interpretation [6,7,8,9,10,11,12,13,14,15,16,17,18,19,20,21,22,23,24,25,26,27,28,29,30,31] which can be addressed by automated scanning systems [11,32] that allow for a ‘semi-quantitative‘ assessment and thus for the estimation of antibody levels (titers). Due to the lack of correlation between LIA results, IP and IIM phenotypes, Mecoli et al. proposed modified cut-offs for the different MSA [12]. This approach appeared to reduce but not to abolish the discrepancy between anti-synthetase antibodies and ASS [33]. In this context it is important to mention that a recent survey indicated that despite concerns about performance of MSA testing, more than 80% of survey participants shared that the MSA testing results influence clinical decision making [6].

Another aspect towards better standardization of MSA assays is the access to proficiency testing initiatives, as performed for many diagnostic tests. All those needs depend on the availability of control material. Close collaboration between research networks [34], patient groups and kit manufacturers is required to supply serum samples for calibration and quality control. Since it can be challenging to obtain large volume bulk samples, pooling of patient samples or the generation of human monoclonal antibodies could provide viable alternatives [35,36]. 

Based on the promising performance characteristics of a novel PMAT system for the detection of MSA observed in previous studies [20,21,22], we aimed to validate those findings. In the current study, the prevalence of all MSA was within the expected range extracted from the published literature [37]. 

All markers exhibited very high specificity, ranging from 98.5% (MDA5) to 100%. More specifically, anti-Jo-1, anti-PL-7, anti-EJ, anti-Mi-2 and anti-SAE antibodies all showed 100% disease specificity. For anti-SRP antibodies, it has recently been shown that the clinical specificity is lower than originally assumed, especially when not accompanied with the typical IIF pattern [38], which appears in line with our observation as we also found anti-SRP antibodies outside the IIM spectrum.

Most of the control samples that tested positive had low values with very few exceptions. One SLE patient with PL-12 antibodies was also very high positivity for RNP (by PMAT connective tissue disease panel; data not shown here). It is possible that the anti-PL-12 antibody positive patient had features that support SLE/IIM overlap syndrome [39]. One ID patient with anti-MDA5 antibodies and one RA patient with anti-SRP antibodies also belonged to this very minor subset.

Interestingly, MDA5 showed the highest prevalence in the control population, including two SLE patients and one individual with infectious disease. Overlap cases of MDA5 associated CADM and SLE have been previously described [40]. Whether the reactivity in the ID patient represents a pathogenic mechanism as discussed previously deserved further research [41,42]. Besides in IIM, anti-NXP-2 antibodies have been described in a patient with SLE and neurological complications [43].

As stated above, anti-Mi-2 antibodies showed relatively low agreement between IP and LIA in previous studies [10,12,20,21,37,44]. This might be related to the different antigens used for the detection (Mi-2 alpha, beta or both). In one of these studies, the new PMAT technology was assessed for the detection of anti-Mi-2 antibodies in a Mexican myositis cohort [20]. It was concluded that the sensitivity for the PMAT anti-Mi-2 antibodies is sufficient with high specificity for DM. In our cohort the prevalence of anti-Mi-2ß antibodies is in line with previous studies [37], and we did not observe any reactivity to Mi-2ß in the controls.

Not many studies have analyzed the combined performance of MSA as an aid in the diagnosis of IIM. In a study by Ghiradello et al. [32], a LIA in comparison to in-house methods was evaluated for overall sensitivity and specificity. For the LIA, the sensitivity was 38%, with a specificity of 92%. In-house testing provided values of 43% and 95%, respectively. The same specificity (95%), but much lower sensitivity (19.9%), was found by Montagnese et al. [45]. Lastly, Platteel et al. [46] found a sensitivity of 47.1% with a specificity of 82.5%. All of the studies mentioned above used the same commercial LIA. In the present study using PMAT, we observed 68.2% (95% confidence interval 62.3–73.5%) sensitivity and 94.0% (95% CI 89.8–96.5) specificity, and a diagnostic odds ratio for IIM of 33.8. Visualization of the post-test probability as a function of the pre-test probability paired with the test result might aid in the assessment of IIM patients as demonstrated for the interpretation of antibody results in celiac disease [47].

Low sensitivity of certain MSA (PL-7, PL-12, NXP2, SAE, SRP) as an aid in the diagnosis of IIM as an entity might not provide much value (Figure 5). However, in the context of IIM subtypes, these markers add significant value both to diagnosis as well as stratification of the sub-type (MDA5, HMGCR, SRP and PL-12 are such examples). A radar plot (Figure 5) depicts this relationship of overlapping and non-overlapping odds ratio values in two different contexts (IIM vs. controls and IIM sub-type vs other IIM cases). Using large datasets combined with artificial intelligence has the potential to further improve the discrimination of IIM and other conditions [48]. This concept is in-line with the study by Montagnese et al. [45] which combined demographic, clinical, and laboratory features in two multivariable logistic regression models. Model 1 was developed to differentiate IIM patients from individuals with no neuromuscular disease, while model 2 was designed to differentiate IIM from non-inflammatory myopathies. In ROC analysis, model 1 (including age, CK level, weakness at onset, EMG and any MSA) showed good discrimination between the two groups (AUC = 0.95). In contrast, limited discrimination was found for model 2 (AUC = 0.80) (including age, myalgia at onset, EMG, CK level).

In line with this concept, we generated composite scores (MyoScores) based on different combinations of markers. The combined score for ASS (Jo-1, PL-7, PL-12 and EJ) showed very high AUC in ROC analysis (0.96), even in the absence of other synthetases (e.g., OJ, HS, Zo), which might represent a promising approach to aid in the diagnosis of ASS [49]. OJ was not included in the study since it represents a very challenging antigenic target. Unlike many of the other MSA, anti-OJ antibodies are directed against a large macro-molecular complex (50) that is difficult to represent with a single antigen. As a matter of fact, 0/14 anti-OJ positive samples were detected using the most widely used LIA [6] in a comparison study with IP [10]. However, significant promise arose from a recent study showing that two components of the OJ complex [namely isoleucyl-tRNA synthetase (IARS) and lysyl-tRNA synthetase (KARS)], are likely sufficient for the effective detection of anti-OJ antibodies [50], a finding that yet has to be verified and transferred to the clinical setting. Antibodies to additional synthetases, once fully characterized [51], might help to further improve such models. Similarly, the model based on HMGCR and SRP shows promising discrimination between IMNM and other forms of IIM. Further studies are needed to further validate the utility of the MyoScores.

## 5. Conclusions

In summary, our study provides additional evidence that MSA carry significant value within their myositis phenotypes. In addition, the use of simple composite scores might help clinicians in the interpretation of MSA results.

## Figures and Tables

**Figure 1 diagnostics-11-02246-f001:**
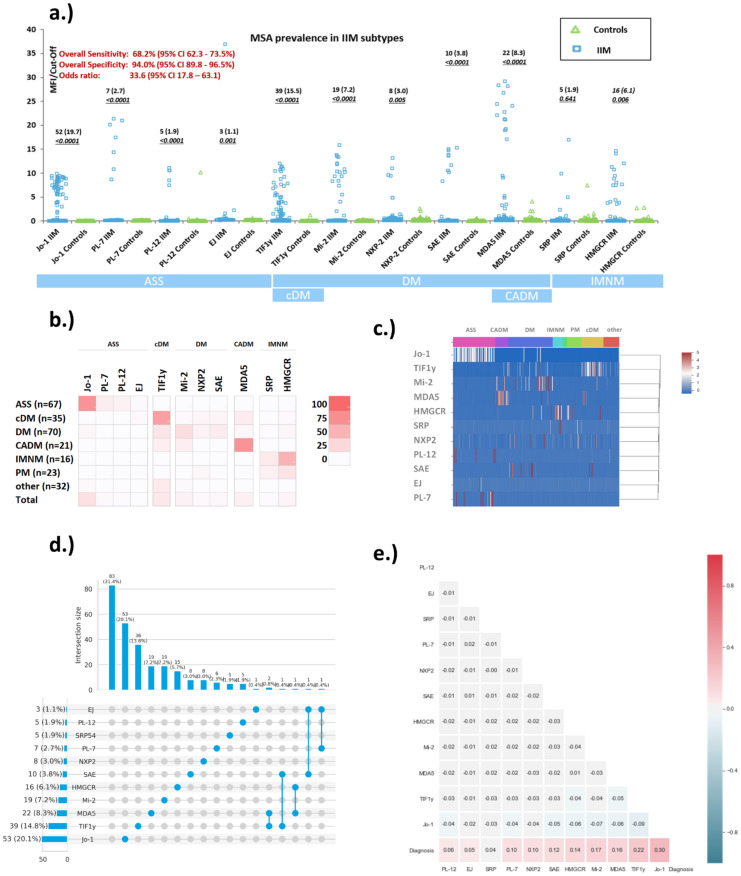
Reactivity of myositis specific antibodies in idiopathic inflammatory myopathy (IIM) patients and controls. (**a**) Scatter plot showing the levels of myositis specific antibodies in different myositis phenotypes as well as in controls. The results derived from particle based multi-analyte technology (PMAT) were normalized by dividing the median fluorescence intensity (MFI) value of the samples by the median fluorescence intensity (MFI) cut-off. (**b**) illustrates the prevalence (%) of the individual MSA in IIM subtypes. (**c**) shows a cluster diagram of MSA. (**d**) Co-existence and correlation of myositis specific antibodies (MSA) for idiopathic inflammatory myopathies (IIM) is depicted using ‘UpSet plot’. As reported in the literature, the majority of MSA are mutually exclusive. (**e**) The correlation diagram shows the relationship between the different markers and the diagnosis. Abbreviations: SRP = signal recognition particle; TIFγ = transcriptional intermediary factor 1 gamma; MDA5 = Melanoma differentiation-associated protein 5; NXP2 = nuclear matrix protein 2; SAE = small ubiquitin-like modifier activating enzyme.

**Figure 2 diagnostics-11-02246-f002:**
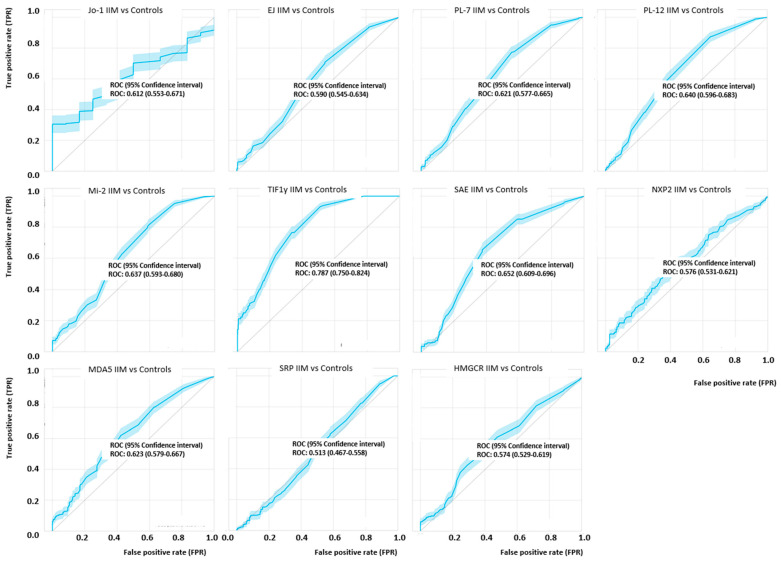
Receiver operating characteristic (ROC) analyses of myositis specific antibodies in idiopathic inflammatory myopathy (IIM) patients and controls. The ROC curves show variable discrimination between IIM and controls with significant discrimination only for Jo-1 and TIF1γ. Antibodies such as anti-EJ, anti-Mi-2, anti-MDA5 and anti-HMGCR showed good discrimination only in the low sensitivity part of the ROC curve which is linked to the different phenotypes combined in the analyses. The area under the curve (AUC) values are summarized in the figure.

**Figure 3 diagnostics-11-02246-f003:**
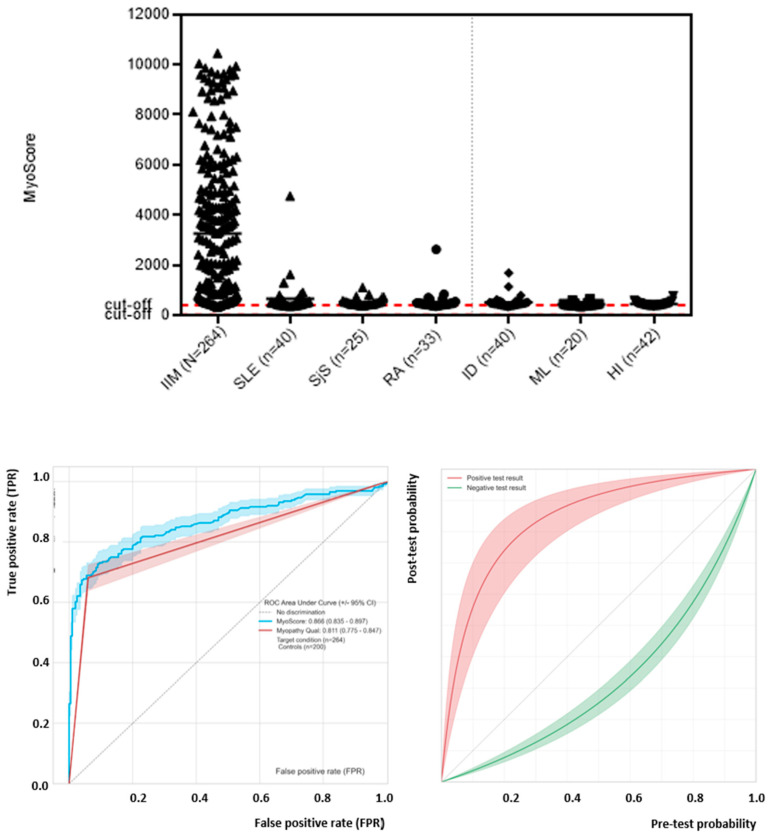
Performance of the MyoScore for idiopathic inflammatory myopathy (IIM). Receiver operating characteristic (ROC) analysis showed good discrimination between IIM patients and controls with an area under the curve (AUC) of 0.87. The area under the curve (AUC) was found as 0.87 (0.82–0.90). Precision recall curve shows high AUC. Pre-test and post-test probability curve illustrates impact of test results (positive and negative) derived from the composite score on the probability for IIM.

**Figure 4 diagnostics-11-02246-f004:**
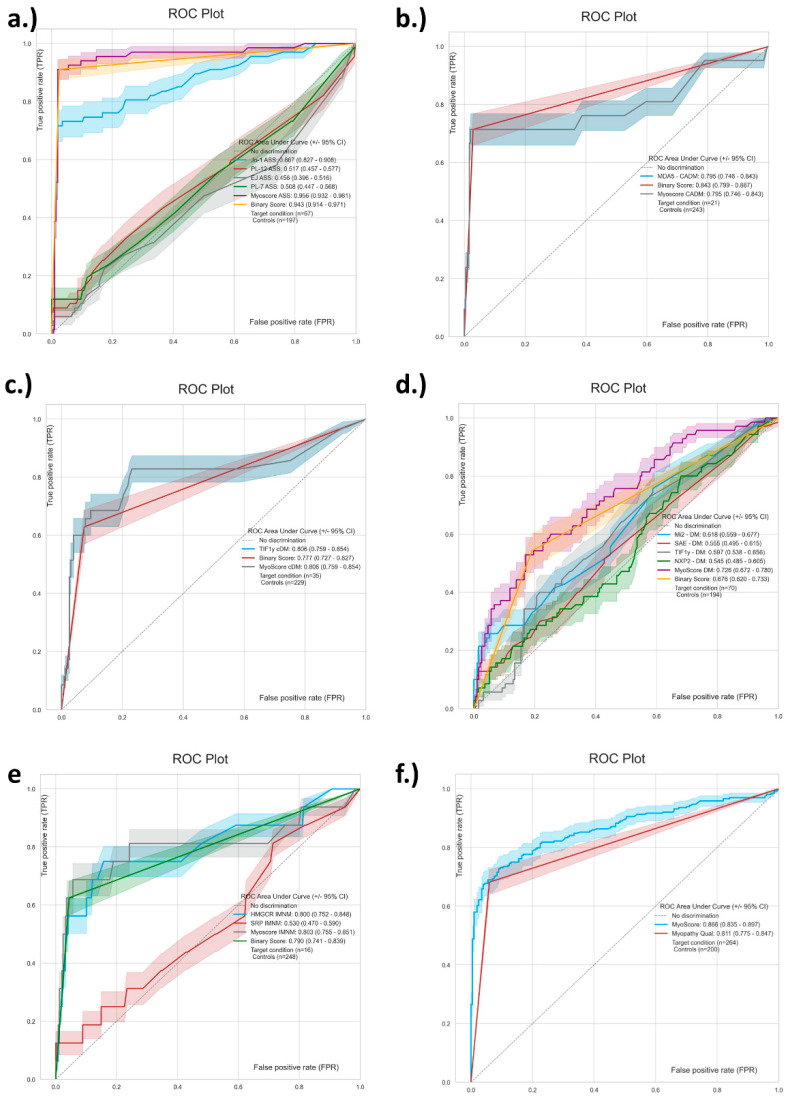
Receiver operating characteristic (ROC) analyses of scores to differentiate myositis phenotypes. The ROC curves showed variable discrimination between the different idiopathic inflammatory myopathy (IIM) subtypes. (**a**) shows the discrimination of antibodies to Jo-1, PL-7, PL-12 and EJ combined in a score to identify anti-synthetase syndrome (ASS) patients. Panel (**b**,**c**) display the curve for MDA5 and clinically amyopathic dermatomyositis and for TIF1γ for cancer associated dermatomyositis. Panel (**d**) visualizes the combination of Mi-2, SAE, TIF1γ, NXP2 in a score for dermatomyositis. In (**e**) the combination of SRP and HMGCR for immune-mediated necrotizing myopathy (IMNM) is displayed. Panel (**f**) shows the total score for IIM. The area under the curve (AUC) values are shown in the individual panels.

**Figure 5 diagnostics-11-02246-f005:**
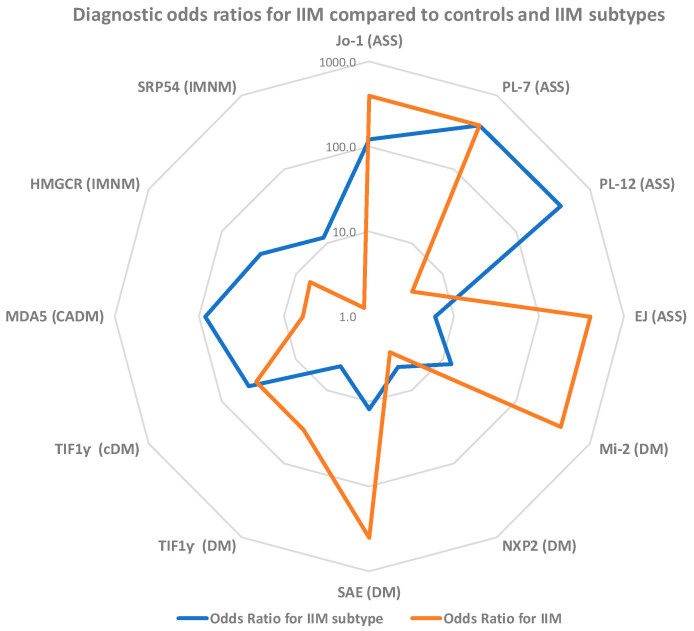
Odds ratio plot of myositis specific antibodies (MSA) in diagnosis and sub-setting of idiopathic inflammatory myopathies (IIM). For markers with very high (infinity) odds, ratios were replaced by a nominally high value of 400 to depict the relative utility of various MSA in the diagnosis of IIM and IIM subtypes. TIF1γ is represented two times (DM and cDM) for IIM; axis for the plot is in log scale. Abbreviations: ASS = Antisynthetase syndrome; DM = dermatomyositis; cDM = cancer associated DM; CADM = clinically amyopathic dermatomyositis; IMNM = immune mediated necrotizing myopathies.

**Table 1 diagnostics-11-02246-t001:** Prevalence of autoantibodies in different disease groups.

IIM MSA	Sensitivity n pos, % pos (95% CI)	Specificity n pos, % pos (95% CI	Odds Ratio (95% CI)
Jo-1	52/264 19.7 (15.3–24.9)	0/200 100.0 (98.1–100.0)	98.1 (6.0–1600.3)
PL-7	7/264 2.7 (1.3–5.4)	0/200 100.0 (98.1–100.0)	27.2 (0.3–2329.2)
PL-12	5/264 1.9 (0.8–4.4)	1/200 99.5 (97.2–99.9)	3.8 (0.6–25.0)
EJ	3/264 1.1 (0.4–3.3)	0/200 100.0 (98.1–100.0)	11.5 (0.1–1066.4)
Mi-2	19/264 7.2 (4.7–11.0)	0/200 100.0 (98.1–100.0)	77.6 (0.9–6377.9)
NXP2	8/264 3.0 (1.5–5.9)	2/200 99.0 (96.4–99.7)	3.1 (0.7–13.0)
SAE	10/264 3.8 (2.1–6.8)	0/200 100.0 (98.1–100.0)	39.4 (0.5–3304.9)
TIF1γ	39/264 15.5 (11.0–19.6)	1/200 99.5 (97.2–99.9)	34.5 (5.9–200.6)
MDA5	22/264 8.3 (5.6–12.3)	3/200 98.5 (95.7–99.5)	6.0 (1.9–19.0)
HMGCR	16/264 6.1 (1.6–23.5)	2/200 99.0 (96.4–99.7)	6.4 (1.6–25.2)
SRP	5/264 1.9 (0.8–4.4)	3/200 98.5 (95.7–99.5)	1.3 (0.3–4.9)

Abbreviations: IIM = idiopathic inflammatory myopathies; MSA = myositis-specific antibodies.

**Table 2 diagnostics-11-02246-t002:** Prevalence of autoantibodies in myositis subsets.

MSA IIM Subtype	Sensitivity IIM Subtype n pos, % ( 95% CI)	Specificity IIM Subtype n pos, % (95% CI)	Odds Ratio (95% CI)
Jo-1 (ASS)	48/67 71.6 (59.9–81.0)	4/197 98.0 (94.9–99.2)	121.9 (40.8–360.0)
PL-7 (ASS)	7/67 10.4 (5.2–20.0)	0/197 100.0 (98.1–100.0)	45.9 (2.6–821.9)
PL-12 (ASS)	5/67 7.5 (3.2–16.3)	0/197 100.0, 98.1–100.0	79.4 (0.9–6999.1)
EJ (ASS)	2/67 3.0 (0.8–10.2)	1/197 99.5 (97.2–99.9)	6.0 (0.7–46.8)
Mi-2 (DM)	15/70 21.4 (13.4–32.4)	4/194 97.9 (94.8–99.2)	13.0 (4.3–38.7)
NXP2 (DM)	5/70 7.1 (3.1–15.7)	3/194 98.5 (95.6–99.5)	4.9 (1.3–19.1)
SAE (DM)	8/70 11.4 (5.9–21.0)	2/194 99.0 (96.3–99.7)	12.4 (2.9–52.9)
TIF1γ (DM)	11/70 15.7 (9.0- 26.0)	28/194 85.6 (79.9–89.8)	4.8 (1.7–13.0)
TIF1γ (cDM )	22/35 62.9 (46.3–76.8)	17/229 92.6 (88.4–95.3)	43.2 (15.1–122.8)
MDA5 (CADM)	15/21 71.4 (50.0–86.2)	7/243 97.1 (94.2–98.6)	84.3 (25.7–277.5)
HMGCR (IMNM)	8/16 50.0 (28.0–72.0)	8/248 96.8 (93.8–98.4)	30.0 (9.2–98.2)
SRP (IMNM)	2/16 12.5 (35.0–36.0)	3/248 98.8 (96.5–99.6)	11.7 (2.15–64.41)

Abbreviations: ASS = antisynthetase syndrome; CADM = clinically amyopathic dermatomyositis; cDM = cancer-associated dermatomyositis; IMNM = immune-mediated necrotizing myopathy; MDA5 = Melanoma differentiation-associated protein 5; NXP2 = nuclear matrix protein 2; SAE = small ubiquitin-like modifier activating enzyme; SRP = signal recognition particle; TIF1γ = transcriptional intermediary factor 1 gamma.

## Data Availability

Data will be made available after reasonable request to corresponding author.

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
