# Peer review of "Profiling of Myositis Specific Antibodies and Composite Scores as an Aid in the Differential Diagnosis of Autoimmune Myopathies"

_diagnostics, 2021, doi:10.3390/diagnostics11122246_

Round 1

Reviewer 1 Report

The manuscript is very interesting and useful for clinicians in aiding the differential diagnosis among different IIMs.

Minor concerns regard the style and many typographical mistakes.

For example:

Please indicate the n.4 affiliation for the Author MTS Martinez

In abstract, line 20, please remove ; at the end of )

In material and methods, line 63, please correct medicated

In material and methods, at lines 78-79, 93, 96, and 116, please extend the abbreviations of MSA, ORs, and LR+ at first use (and also extend AUC).

In Table 1, please add /200 in PL-12 & TIF1gamma lines, in the specificity column for both

In Fig.1a, please reposition the label of y-axis on the left of this axis. In d, please explain SRP54. In the legend, please clarify the numbers showed inside this Figure and be consistent with the tenses of verbs.  

In Fig. 2, 3, and 4, please magnify the font used for all the data inside each panel.

Please, erase . at line 132.

In line 179, there is a lack of (

Fig. 4, please add .) in panel e.

All lettering in Legends should be reported in lower case, as in Figures.

In Line 216-217, please rephrase this difficult sentence.

Author Response

Reviewer 1

The manuscript is very interesting and useful for clinicians in aiding the differential diagnosis among different IIMs.

Minor concerns regard the style and many typographical mistakes.

For example:

Please indicate the n.4 affiliation for the Author MTS Martinez

Thank you, affiliation has been removed

In abstract, line 20, please remove ; at the end of )

Thank you for pointing this out to us. Has been corrected

In material and methods, line 63, please correct medicated

Thank you for pointing this out to us. Has been corrected

In material and methods, at lines 78-79, 93, 96, and 116, please extend the abbreviations of MSA, ORs, and LR+ at first use (and also extend AUC).

Abbreviations have been introduced as requested

In Table 1, please add /200 in PL-12 & TIF1gamma lines, in the specificity column for both

The missing information has been added to the table

In Fig.1a, please reposition the label of y-axis on the left of this axis. In d, please explain SRP54. In the legend, please clarify the numbers showed inside this Figure and be consistent with the tenses of verbs.  

Figure has been changed accordingly. SRP54 was changed to SRP

In Fig. 2, 3, and 4, please magnify the font used for all the data inside each panel.

Thank you. Font sizes have been adjusted

Please, erase . at line 132.

Thank you. Edit has been made

In line 179, there is a lack of (

Thank you. Edit has been made

Fig. 4, please add .) in panel e.

Thank you. Edit has been made

All lettering in Legends should be reported in lower case, as in Figures.

Lettering has been adjusted

In Line 216-217, please rephrase this difficult sentence.

Thank you. Sentence has been reworded

Reviewer 2 Report

It is a fascinating and original article, which allows a better understanding of the pathophysiology of myositis.
A large number of patients analyzed makes it different from what has been published to date.
I found the way of describing the information with the tables to be excellent, as it summarizes the complexity of the information obtained in very illustrative and straightforward graphs.
I congratulate the authors for this great work.

Author Response

Thank you for reviewing our paper and the feedback